# Effect of Environmental Enrichment and Herbal Compounds-Supplemented Diet on Pig Carcass, Meat Quality Traits, and Consumers’ Acceptability and Preference

**DOI:** 10.3390/ani8070118

**Published:** 2018-07-16

**Authors:** Nicolau Casal, Maria Font-i-Furnols, Marina Gispert, Xavier Manteca, Emma Fàbrega

**Affiliations:** 1IRTA, Veïnat de Sies s/n, 17121 Monells, Spain; nicolaucasal@gmail.com; 2Department of Animal and Food Science, School of Veterinary Science, Universitat Autònoma de Barcelona, 08193 Bellaterra (Barcelona), Spain; xavier.manteca@uab.es; 3IRTA, Finca Camps i Armet s/n, 17121 Monells, Spain; maria.font@irta.cat (M.F.-i.-F.); marina.gispert@irta.cat (M.G.)

**Keywords:** pork quality, environmental enrichment, herbal compounds, conjoint analysis, sensory analysis

## Abstract

**Simple Summary:**

Interest in animal welfare has increased and been considered a relevant attribute for the concept of the ethical quality of pig products. The present study suggests that consumers would appreciate particular improvements on animal welfare such as providing environmental enrichment or herbal compounds. The provision of those welfare improvement strategies did not have a significant effect on technological carcass and meat quality parameters. However, the strategies used in this study that can increase animal welfare in production systems were valued by consumers.

**Abstract:**

Animal welfare can be considered an ethical attribute of product quality, but consumers should appreciate its added value. The aim of this study was to evaluate consumer’s acceptability, preference, and the meat and carcass quality of pigs reared with two stress-reducing strategies: supplementation of an herbal compound (HC) containing *Valeriana officinalis* and *Passiflora incarnata*, and environmental enrichment (EE) by the provision of hemp ropes, sawdust, and rubber balls. A total of 56 pigs were divided in four treatments in two pens of seven pigs per treatment (2 × 2 factorial design). Meat and carcass quality were evaluated. Consumer’s acceptability and preference were analysed with a sensory test and a conjoint analysis in 110 consumers. Before slaughter, control pigs (no EE and no HC) presented lower live weight compared with other treatments (*p* = 0.0009). Although acceptance was the same for all of the treatments, consumers preferred systems aiming to increase pig welfare. The most important factor was production system, with a preference for those improving welfare, followed by feeding system, with a preference for those with natural herbs supplementation. Although price was the least important factor, a segment of consumers showed a clear preference for lower prices. These results suggest that welfare improvements could be appreciated by particular consumer segments.

## 1. Introduction

Intensive pig production conditions subject individual pigs to different stressors that may negatively impact on their well-being. Moreover, it has been argued [1] that increasing animal welfare has a positive impact on product quality, food safety, and the healthiness of consumers. As a consequence, an increase in the demand for more welfare-friendly food products has been observed [2]. Consumer behaviour may be influenced both by intrinsic characteristics of pork such as colour, taste, texture, and odour, and some extrinsic characteristics such as price, food safety, origin, and information on animal production [3]. The same product attributes may be regarded with different consumer attitudes, beliefs, and behaviours. From those attributes, animal welfare is considered to be one of the most important for certain pork consumers [4]. However, the willingness to pay more for products with an animal welfare guarantee reported by some consumers does not always translate into a real consumption behaviour. Along that line, the Europeans recognised, according to the 2016 special Eurobarometer, that a price premium is justified for animal welfare-friendly products, with 59% of respondents prepared to pay more for products sourced from animal welfare-friendly production. However, when evaluating the answers at a national level, those positive responses ranged from 93% to 22%. Consumers that purchase animal-friendly products consider that having access to information regarding the welfare issues and being able to trust that information provided in the labels are important factors in their decision-making process. In that sense, a considerable sector of the consumers indicate a lack of information about production systems and market transparency [5]. According to Eurobarometer 2016 [4], there has been a shift in opinion from the last survey, with a mean 47% of Europeans considering there is not a sufficient choice of animal welfare-friendly products available in shops and supermarkets, versus 38% considering them to be sufficient. Besides, most consumers would appreciate having access to more traceable and transparent general information, although they would not require very specific details regarding the way the animals were raised [6]. In contrast, some authors point out that the industry interest for animal welfare is basically economic [7]. 

Increase in animal welfare in pig production systems can be achieved by implementing different strategies. For this particular study, the two management strategies selected were: the provision of environmental enrichment and supplementation with herbal compounds with sedative properties. Both strategies have been previously found to reduce stress in pigs [8,9]. According to European Union legislation (Council directive 2008/120/EC), the provision of enrichment materials is mandatory in order to prevent the behavioural and physiological problems that may arise from rearing pigs in non-enriched conditions. Pigs are highly motivated for rooting and performing exploratory behaviour, the so-called behavioural needs [10]; thus, enrichment materials aim at enhancing some species-specific behavioural patterns such as foraging and at reducing, in turn, pig frustration. 

Previous studies have suggested that the provision of environmental enrichment during rearing may be associated with a stress reduction under pre-slaughter procedures and result in better quality pork [11]. This may be due to a better coping capacity of pigs provided with enrichment, whereas pigs reared in non-enriched environments may show greater “reactivity” in response to stressful stimuli or more anti-social, aggressive, and harmful behaviours [10]. Along that line, the provision of enrichment has been linked to beneficial effects in meat traits such as water-holding capacity or tenderness [12].

The main active components of the herbal compound supplemented (sedafit^®^, Phytosynthèse, Saint-Bonnet de Rochefort, France) were *Valeriana officinalis* and *Passiflora incarnata*. In both plants, there are bioactive compounds with sedative or relaxing properties that use the interaction of valerian acid with the γ-Amino butyric acid receptors type A (GABAA) as action mechanism [13]. Furthermore, other bioactive compounds have been detected in *Passiflora incarnata* such as maltol, flavonoids, indole alkaloids, and cyanogenic glycosides, although there is no general agreement on which one is the most important with regard to the sedative properties. The effect of flavonoids, which is one of the more studied compounds, is similar to valerian acid: the membrane permeability is increased by means of the modulation of GABAA. The authors are not aware of any studies reporting the effects of those herbal compounds on carcass and meat quality in pigs. The hypothesis under study would be that no detrimental effect in terms of acceptance of meat would be expected, and that some benefits from reduction on stress could be expected with regard to some parameters such as pH.

The aims of the present study were to: (1) evaluate carcass and meat quality characteristics of pork produced using two strategies aiming to reduce the stress on farm: supplementation of an herbal compound with sedative properties, and the addition of enrichment material to the pens; (2) evaluate consumers’ sensory acceptability of meat from these different strategies; and (3) determine the effect of three factors (feeding supplementation, production system, and price) on consumer’s purchasing intention of pork.

## 2. Materials and Methods

### 2.1. Animal and Housing Conditions

The housing, husbandry, and use of the animals for the procedures described in this manuscript were carried out according to the European legislation for animals used for scientific purposes (Directive 2010/63/EU of the EU Parliament and of the Council of 22 September 2010 on the protection of animals used for scientific purposes). The project, including this experimental procedure, was approved by IRTA’s (Institute of Agrifood Research and Technology) Ethical Committee. According to Directive 2010/63/EU, “*any restrictions on the extent to which an animal can satisfy its physiological and ethological needs have to be kept to a minimum*”. In the present experiment, no environmental enrichment was provided to the control group, because the authors aimed at investigating the effects of the provision of enrichment on behavioural and physiological performance, and meat and carcass quality traits. For that reason, the Ethical Committee had to decide according to Article 38 of European Union (EU) legislation on whether “*the harm to the animals in terms of suffering, pain, and distress is justified by the expected outcome taking into account ethical considerations, and may ultimately benefit human beings, animals, or the environment*”. The Ethical Committee provided its approval to the procedure considering that the data obtained regarding the provision of enrichment would yield important outcomes to be implemented on commercial farms, for which, according to EU legislation (EU Council directive on pig welfare 2008/120/EC), enrichment material is mandatory. In order to minimise the possible effects of non-provision of enrichment to the control group, other risk factors that can influence the fulfilment of pig behavioral needs and, as a consequence, the occurrence of tail biting, were cared for: space allowance per pig was 1.95 m^2^/animal; climatic conditions were automatically controlled, and daily supervision was carried out according to the Ethical Committee recommendations, in order to apply correction measures if necessary.

Fifty-six entire male pigs [(Landrace × Large white) × Pietrain] were used in the present experiment. The average live weight of pigs at arrival at the experimental farm was 25 ± 0.27 kg (mean ± SE) with 10 weeks of age. Pigs were kept together until the experiment started at the age of 16 weeks (and average weight of 49.8 ± 0.56 kg). Pigs were randomly assigned to one of the four different treatments (14 pigs/treatment in total): (a) pigs supplemented with both enrichment material and herbal compound (EEHC); (b) pigs supplemented with enrichment material (EE); (c) pigs supplemented with the herbal compound (HC); and d) control group (CG). Two identical rooms were used; thus, there was one replicate of seven pigs per treatment in each of the two rooms.

Pens had the same size (13.67 m^2^), and visual contact between pigs of adjacent pens was allowed by using metallic fences. The space availability per pig was 1.95 m^2^, and food and water were provided *ad libitum* by means of one drinker and two hoppers. Natural hemp ropes, sawdust, and rubber balls were provided altogether during all of the experiments as enrichment materials in the EE treatment. More precisely, the hemp ropes (two) that were hung in the walls of the environmentally enriched pens had 80 cm of length and were replaced when the length was lower than 30 cm. The provision of sawdust required that the 1/3 of the slats of EE and EEHC pens were covered with polypropylene sticks (Click-in^®^, Rotenca, Agramunt, Spain). Two full buckets (50 × 60 cm) were added every two days. The EE treatment also included a single rubber ball with a diameter of 15 cm provided at the beginning of the experiment and cleaned regularly to avoid soiling. The conventional environment consisted of a fully slatted floor with no enrichment material provided. The herbal compound used (Sedafit ESC, Phytosynthèse, Saint-Bonnet de Rochefort, France) contained *Valeriana officinalis* and *Passiflora incarnata,* and was manually added to the food concentrate of the HC and EEHC treatments in a concentration of 2000 mg/kg. Pigs were fed *ad libitum* following a phase feeding regime with a commercial concentrate (Esporc, Riudarenes, Spain, containing 17.02% crude protein and 3.91 Kcal EM at mid fattening).

As part of a broader study, behavioural observations, body weights, and skin lesions were periodically registered, and blood, saliva, and hair samples were collected [14]. As presented in Casal et al. [14]., body weight at 16 weeks was not significantly different between treatments (50.22 ± 1.17; 49.61 ± 1.05; 49.64 ± 1.26; 49.81 ± 1.22, for CG, HC, EE, and EEHC, respectively), whereas at 24 weeks of age, the control group presented a significantly lower weight compared with the EE, HC, and EEHC pigs (104.5; 110.35; 111.79 or 112.35 SEM = 0.69, respectively, *p* < 0.05, for more details see Casal et al. [14].

### 2.2. Carcass Quality Measurements

All of the animals were slaughtered in a commercial abattoir at the age of 24 weeks with a mean live weight of 109.77 ± 1.35 kg. All of the pigs were gas stunned before sticking. Pigs were fasted for 12 h on a farm, transport to the slaughterhouse took 30 min, and they were kept in lairage for 2 h before slaughter. Pigs were handled using gentle handling to load and unload and avoiding the mixing of unfamiliar pigs on the truck. Carcass weight was measured 45 min *post mortem* (p.m.), and carcass yield was calculated by dividing carcass weight and live body weight. Furthermore, backfat (LR3/4FOM) and muscle thicknesses (MFOM) were measured between the third and fourth last ribs, at 6 cm from the midline, using a Fat-O-Meat’er probe (Frontmatec A/S, Herlev, Denmark). The Spanish official equation was used to calculate the lean percentage from the backfat and muscle thickness measurements. A ruler was used 24 h *post mortem* to take different measures on the left carcass of each animal: (a) minimum fat and (b) skin thickness (perpendicular to the skin) over the *gluteus medius* muscle (MLOIN). A tape was used to measure loin (from the atlas to the first lumbar vertebra) and carcass length (the first rib to the anterior edge of the pubic symphysis).

### 2.3. Meat Quality Measurements

A Crison portable meter (Crison, Barcelona, Spain) equipped with a Xerolyte electrode and a Pork Quality Meater (PQM-I INTEK Gmbh, Aichach, Germany) were used 24 h p.m. to measure muscle pH and electrical conductivity in the muscle *Longissimus thoracis* (LT) at the last rib level, respectively. At the same moment, three trained observers evaluated the colour of the exposed cut surface of the LT muscle at the last rib level using the Japanese scale colour (1: very pale to 6: very dark) [15]. Luminosity L*, tendency to red a*, and tendency to yellow b* (colour parameters on the CIELab space) were obtained with a Minolta Chromometer (CR-400, Minolta Inc., Osaka, Japan) [16]. Drip loss was also evaluated from the LT according to the methodology described by Rasmussen and Andersson [17]. Three technicians used the National Pork Production Council pattern [18] to evaluate marbling on the LT muscle at the last rib level. Moreover, LT samples were also taken and frozen at −20 °C to analyse intramuscular fat and texture. Intramuscular fat content (GRINLD) was analysed with an infrared FoodScan equipment (FOSS analytical, Hillerød, Denmark) at wavelengths between 850–1050 nm. Samples were thawed for 24 h at 2 °C, and ground using a Robot-Coupe Blixer 3 blender (Seysant Atlantic S.L., Soria, Spain). Shear force was determined in LT samples from the second to the third last rib level. LT was thawed 24 h at 2 °C. Then, chops were cooked in a pre-heated oven to 200 °C (Spider 5, Novosir, Spain) until the internal temperature reached 75 °C. After 2 h at room temperature, pieces of 3 × 1.5 × 1.5 cm^3^ were collected for the analysis. Then, pieces were sheared using an Alliance RT/5 texture analyser (MTS Systems Corp., Eden Prairie, MN, USA) equipped with a Warner–Bratzler blade with crosshead speed set at 2 mm/s, and peak load (kg) was recorded.

### 2.4. Consumers’ Study

The consumers’ study was carried out in Barcelona with a total of 110 consumers. The selection of consumers was made so that the sample was representative of the Spanish population according to gender and age, although a slight bias in gender was finally obtained (Table 1). A total of 11 sessions of 10 consumers each were conducted.

First, a sensory analysis acceptability test was performed to determine if consumers were able to find differences in the acceptability of loins depending on the treatment. Loin sections that were 1.5 cm thick were cooked in a pre-heated oven at 200 °C until the internal temperature reached 76 °C, which is the temperature recommended for sensory evaluations [19]. Once cooked, each slice was divided in four pieces of 1.5 cm-thick (perpendicular to the subcutaneous fat), and were covered with aluminium foil, codified with a three-digit code, and kept warm until being served to the consumers. A sample of each treatment was offered to each consumer so that all four samples were evaluated in blind conditions. Samples were distributed to the consumers monadically and following a design to avoid the first sample and carry over effect [20]. Consumers evaluated the overall acceptability/liking, tenderness, smell liking, and flavour liking according to a nine-point scale (from 1: ‘dislike very much’/‘very hard’, to 9: ‘like very much’/‘very tender’) without the intermediate level (5: ‘neither like nor dislike’).

Second, a conjoint analysis was used to undertake a preference study. Feeding, production system, and price were the three factors that were evaluated. In that sense, two levels were considered for feeding: (a) conventional feed, and (b) conventional feed supplemented with natural herbs with a sedative effect. The two levels evaluated for production system were: (a) conventional farming system and (b) conventional farming system with improvements for the animal welfare. Finally, three levels were considered for price: 3 €/kg, 5 €/kg, and 7 €/kg. A full design was used; thus, 12 different profiles were obtained from all the possible different combinations of the levels of each factor. Each profile was represented in a card, and consumers were asked to look at the cards carefully and rank them according to their purchase intention from 1 (less preferred) to 12 (most preferred). Four consumers were discarded for this analysis, because of non-properly recorded answers; thus, data of the conjoint analysis is based on 106 subjects. 

### 2.5. Statistical Analysis

Statistical Analysis Software (SAS version 9.2; SAS institute Inc., Cary, NC, USA) was used to conduct the statistical analysis. Significance was established at *p* < 0.05 for all of the analyses, while a tendency was considered between *p* > 0.05 and *p* < 0.1.

For carcass and meat quality parameters traits, general linear models (MIXED, model allowing for fixed and random effects, procedure) were applied. The fixed effects taken into consideration were use or not of herbal compound, environmental enrichment, and their interaction, including carcass weight as covariate in the carcass quality variables (models are provided at Appendix A). Tukey’s test was applied to test for differences between least square means of fixed effects. 

For the consumer’s conjoint analysis, a Ward method was used in an agglomerative hierarchical cluster analysis (PROC CLUSTER) to determine different segments of consumers according to the ratings given to the cards. Clusters were selected from the dendogram obtained (see Appendix A), trying to maintain as much as homogeneity within each cluster but at the same time, maximising the heterogeneity between clusters. Conjoint analysis was analysed for all of the consumers as a pool and by cluster using the TRANSREG, transformation regression model, procedure. Demographic differences among different clusters were analysed using the GraphPad QuickCalcs website for statistics by means of the chi-square test two by two (http://graphpad.com/quickcalcs/chisquared1/).

The MIXED procedure by cluster and for all of the consumers as a pool was used to analyse the sensory attributes of meat. The model included use or not of environmental enrichment and herbal compound, and their interaction as fixed effects, consumer was included as a random effect, and session was included as a blocking effect (model is presented at Appendix A). Tukey’s test was applied to test for differences between the least square means of fixed effects. 

In order to find the relationship among all of the variables, a principal component analysis was performed with the Factor procedure of SAS using varimax rotation. A selection of carcass and meat quality variables, including all of the different aspects of quality and avoiding similar variables were included, as well as acceptability of meat by consumers.

## 3. Results

### 3.1. Body Weight, Meat and Carcass Quality

Results are presented for main factors (Table 2), because no interaction between HC and EE was found. Body weight previous to slaughter was significantly higher both for pigs provided with EE (112.07 ± 2.49 versus 107.57 ± 1.17 kg, *p* = 0.002) and supplemented with HC (111.19 ± 2.39 versus 108.45 ± 3.04 kg, *p* = 0.02, Table 2). Moreover, yellowness (b*) was higher in the pigs with EE (*p* = 0.004). Pigs supplemented with herbal compound tended to present a higher minimum fat thickness over the *gluteus medius* muscle (*p* = 0.060) compared with those non-supplemented (11.10 ± 0.6 mm versus 9.91 ± 0.37 mm, respectively). The rest of the parameters evaluated (carcass weight, carcass yield, last rib backfat, muscle thicknesses, carcass lean meat (%), carcass length (cm), loin length (cm), MLOIN, conformation, pH, ECult, L*, a*, colour EJC (subjective colour using a Japanese colour scale), drip loss, marbling, intramuscular fat%, and shear force (g/cm^2^) were not significantly affected by the treatments (*p* > 0.05).

### 3.2. Consumers’ Study

The sensory characteristics of the meat scored by the consumers revealed no significant differences in any of the characteristics assessed for either pigs supplemented with the herbal compound nor the animals provided with environmental enrichment. Averaged values for overall acceptability, tenderness, odour, and taste were 5.60 ± 0.08, 4.90 ± 0.09, 5.91 ± 0.07, and 5.75 ± 0.08, respectively (*p* > 0.05). The individual study of clusters showed similar results, with no significant differences between treatments (*p* > 0.05).

Table 3 summarises the results of the conjoint analyses. In the pooled study of consumers, production system appeared as the most important factor (45.7% of importance), with a preference for those systems with improvements for the animal welfare as indicated by the higher utility value. The second factor considered by the consumers (34.4% of importance) was the feeding system, with preference for conventional feed supplemented with natural herbs with relaxant properties. The less important factor was the price, being the intermediate price the most preferred.

Consumers from this study were clearly divided in three different clusters (Table 3). Cluster one was composed by 26.4% of the consumers, with price being the most important factor for them (72% of importance), with lower price as the most preferred. Feed supplementation received 14.3% of importance, and production system 13.7%. This cluster presented a higher proportion of men (*p* = 0.01) compared with the total distribution (Table 1).

Cluster two was the biggest, with half of the consumers (51.8%), and the gender, age, and level studies distribution was similar to the overall one. Cluster 2 scored feeding (45.9% of importance), and production system (40.4% of importance) as the most important factors, with a preference for food supplemented with herbs and animal welfare improvements, respectively. Price was the least important attribute for this group (13.7%), and showed a preference for the intermediate price. 

Finally, cluster three (21.7% of consumers) considered price to be the most important factor (48.4% of importance), but in contrast with cluster 1, their preference was for the highest price. Feeding system was also important for this cluster (38.4% of importance), with a preference for conventional feeding. Production system was considered the least important attribute for this group, although they preferred those systems reared with enrichment material. Consumers from this cluster tended to have less university and more secondary studies than the overall population (*p* = 0.07, Table 1).

### 3.3. Relation between Quality Variables

The result of the principal component analysis is presented in Figure 1. The first principal component explained 22.8% of the variance accounted for, and the second one explained 21.2% of this variance. The correlation of the attributes with the first two factors are represented. Additionally, the plot includes the average of the coordinates of the different observations by treatment. It is possible to see that the first principal component is related in its positive part with the acceptability of meat by consumers, the colour parameters, and the drip losses. Thus, this is the axis related to meat quality. The second principal component is mainly related to the carcass quality variables, fat content as the positive part and lean content as the negative part. Logically, higher fat is negatively correlated with higher lean meat content. Furthermore, higher lean meat content in the plot is also related with higher shear force. No relation between carcass and meat quality characteristics was found.

## 4. Discussion

The results presented are part of a broader study aiming to (1) evaluate the effects of the mitigation of chronic stress by means of environmental enrichment and herbal compounds on behaviour and physiological indicators, and (2) assess the effect of those strategies on meat and carcass quality, and consumer’s preference and acceptability in growing pigs and pork. The second aim is discussed below.

### 4.1. Carcass and Meat Quality

Limited effects were found in the carcass and meat quality traits in relation with enrichment material and the herbal compound. The only difference reported in carcass traits was an increased body weight before slaughtering for both pigs supplemented with HC and provided with EE. These differences may be explained by a better growth rate when environmental enrichment and/or herbal compounds were provided, as presented in Casal et al. [14]. Although previous studies with environmental enrichment have reported conflicting results, in general, the provision of straw improves performance [11]. However, when point source objects are provided, these differences are less consistent [21]. Previous studies have related the benefits of providing enrichment with more interaction with feeders and feed consumption [22], or with a reduction of stress, and consequently lower levels of catabolic hormones such as cortisol and catecholamines [23]. Both findings could be underlying better performance records. Along these lines, feed consumption was not recorded in our study, but the levels of cortisol were found to be lower for the enriched pigs, whereas the levels of exploratory behaviour were increased [14]. With regards to herbal compounds provision, results presented in Casal et al. [14] showed lower social interactions and body lesions in supplemented pigs, and it could be argued that some energy could have been diverted from social behaviours to growth.

Pigs raised in enriched environments presented a higher yellowness, although the differences were not relevant because they probably would not be detected by the human eye using the subjective colour evaluation. Furthermore, since yellowness is more important for fat than for lean, and it was measured in the lean, differences in this parameter are not relevant. Therefore, no clear effect of treatment on this particular trait may be inferred from the present results

Our results present no major differences between pigs provided with enrichment or not with regards to carcass and meat quality parameters, which is in agreement with previous studies [24]. However, a better water-binding capacity has been reported in pigs reared in enriched conditions [11,12,25].

### 4.2. Consumer’s Intention to Purchase

Conjoint analyses are run to obtain information about how the attributes of the product analysed can affect liking and/or purchase intention and the relative importance of each attribute. 

The overall conjoint analysis of the present study revealed that production system was the most determinant attribute, followed by feeding with herbal compound supplementation, and price was the least important attribute. A preference for animal-friendly products has also been reported in previous investigations [2]. In contrast, other studies have found that although 77% of European consumers expressed a concern for animal welfare [26], they seemed to be more influenced by other factors such as origin, flavour, price, and meat quality [27]. 

Normally, clusters for consumers can be identified from their behaviour when submitted to a conjoint analysis. The results of the present study may be discussed from the gender composition, with cluster one mainly composed of men, whilst most of the consumers from clusters two and three were women. Previous studies have reported, in line with our findings, a more important influence of meat price in men, with a preference for lower prices [28], whereas women showed a higher concern for animal welfare. The higher implication in household tasks associated with animal care has been provided as a hypothesis to explain the higher preference of women for animal welfare compared to men [1]. 

Cluster two presented the highest interest for animal welfare improvements. Furthermore, this cluster also presented a preference for intermediate prices that could be probably associated with the general perception that cheap products may come from unreliable origins [3], and could also be related with more willingness to pay for animal welfare-friendly products [29]. Price was considered to be the most important attribute for both cluster one and cluster three, but whilst cluster one preferred the cheapest meat, cluster three gave priority to the most expensive. Sasaki et al. [30] found similar results in beef, with divergent clusters showing preference either for the lowest prices or for the highest. On the contrary, investigations from Font-i-Furnols et al. [28] in lamb meat and Mesías et al. [31] in beef meat showed a preference for the lower prices in all of the segments.

Surprisingly, cluster three presented a preference for a conventional production system and for the most expensive meat. A potential explanation is that consumers in this cluster associated more expensive prices with attributes such as better product quality, thus suggesting that this cluster might be more interested in quality aspects than in animal welfare improvements. Another possible hypothesis to explain the results of this cluster may be related somehow with a group defined as the *indifferent* by Dutra et al. [32], who were not interested in the way that pigs are raised. 

The authors would like to acknowledge some limitations in our results. First of all, a higher number of subjects, with a more balanced gender ratio, in the consumers’ study would be required to obtain stronger conclusions. Second, results should be contrasted with those from consumers coming from more rural areas, since urban consumers have been described as having less knowledge in pig production systems but an increased concern for animal welfare [7]. Overall, the present results show the importance of animal welfare for the consumers evaluated. However, a further study dealing with the limitations presented would be interesting to obtain more general conclusions.

### 4.3. Relation between Quality Variables

A principal component analysis (PCA) plot confirmed the low effect of the four treatments in the carcass and meat quality and consumer’s acceptability of the meat. The first axis is mainly related to the meat quality parameters and consumers’ acceptability, and the second axis is mainly related to the carcass quality parameters. It is possible to confirm that higher live weight is related with higher fat content and lower lean meat content because of the deposition of the fat with the weight of the pig [33]. Marbling is not related to consumers’ acceptability scores in accordance with Moeller et al. [34], but it is in disagreement with Font-i-Furnols et al. [35]. Nevertheless, variations in marbling in the present work were not very high, and this can influence the lack of relationship between this characteristic and acceptability scores. Furthermore, no relationship has been identified between carcass and meat quality variables since they are both in the first two axes, which are uncorrelated between them. 

In the conditions of the present experiment, it can be concluded that pork consumers scored animal welfare as an important attribute, with a certain preference for systems aiming to increase the welfare. Women appear to be more concerned about animal welfare, with a lower consideration for price compared to men. Meat and carcass quality were not affected by the provision of environmental enrichment nor herbal compounds, although pigs supplied with herbal compounds, environmental enrichment, or both presented a higher weight compared with the control group.

## Figures and Tables

**Figure 1 animals-08-00118-f001:**
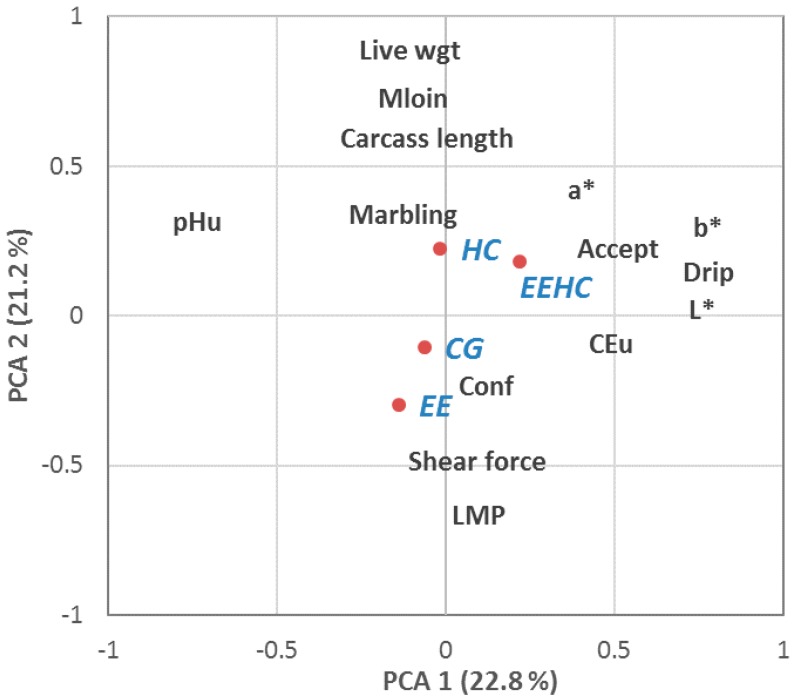
Principal component analysis: correlation of the variables with the principal components and average coordinate of the observation by treatment (CG: not supplied and no enriched environment; HC: supplied with herbal compound; EE: Enriched environment; EEHC: supplied with herbal compound and enriched; Conf: conformation, LMP: lean meat percentage; Accept: acceptability according to consumers’ evaluation). MLOIN: minimum fat over the Gluteus medius. pHuLT: muscle pH (acidity) at Longissimus thoracis 24 h post mortem (p.m.); ECuLT = electrical conductivity (ability to transport electrical charges) measured in the Longissimus thoracis; Marbling NPPC: subjective marbling (intramuscular fat) with pattern from National Pork Producers Council (means of three assessors); L*, a*, b*: objective measure of the colour with the Minolta Chromameter.

**Table 1 animals-08-00118-t001:** Socio-demographic characteristics of the consumers (*n* = 106) ^1^.

Characteristics	Overall	Cluster 1 ^2^	Cluster 2	Cluster 3
Consumers (%)	100	26.41	51.89	21.70
Gender (%)				
Women	62.3	39.3 ^a^	69.1	73.9
Men	37.7	60.7 ^b^	30.9	26.1
Age (%)				
<26	5.7	7.1	3.6	9.1
26–40	30.5	39.3	29.1	22.7
41–55	28.6	35.7	25.5	27.3
56–70	32.4	17.9	40	31.8
>70	2.9	0	1.8	9.1
Finished levels of studies (%)				
Primary school	8.7	3.6	11.3	8.7
Secondary school	55.8	50	50.9	73.9
University	35.6	46.4	37.7	17.4

^1^ Four consumers out of the initial 110 were not considered because information was not provided. ^2^ Different letters within cluster and demographic characteristics indicate significant differences between the overall distribution and the distribution within the cluster.

**Table 2 animals-08-00118-t002:** Least square means of meat and carcass quality traits from pigs reared or not with environmental enrichment and supplemented or not with an herbal compound with sedative properties.

Quality Traits	Housing System (HS) ^b^		Herbal Compound (HC) ^c^		SEM ^a^	*p*-Values ^d^	
	CE	EE	no	yes		HS	HC
Carcass quality traits							
Live weight (kg)	107.57	112.07	108.45	111.19	1.35	0.0002	0.02
Carcass weight (kg)	80.88	82.39	80.89	82.38	1.02	0.37	0.38
Carcass yield (%)	75.18	73.78	74.57	74.38	0.20	0.07	0.58
Last rib backfat (mm)	15.17	15.26	15.30	15.12	0.35	0.86	0.72
Muscle thicknesses (mm)	57.18	58.74	57.68	58.23	0.60	0.14	0.60
Carcass lean meat (%)	61.58	61.79	61.55	61.82	0.30	0.70	0.62
Carcass length (cm)	83.07	82.80	82.90	82.97	0.37	0.56	0.87
Loin length (cm)	84.33	84.15	84.32	84.16	0.37	0.74	0.76
MLOIN	10.30	10.72	11.10	9.91	0.35	0.51	0.06
Conformation	2.72	2.64	2.63	2.73	0.06	0.54	0.50
Meat quality traits							
pHuLT	5.59	5.58	5.59	5.58	0.01	0.70	0.64
ECuLT	4.33	4.40	4.39	4.34	0.08	0.69	0.82
Lightness L*	48.28	49.17	48.52	48.93	0.25	0.11	0.44
Redness a*	6.93	7.00	6.86	7.07	0.11	0.75	0.36
Yellowness b*	1.25	1.72	1.36	1.60	0.10	0.04	0.28
Colour EJC	2.44	2.24	2.35	2.33	0.06	0.13	0.85
Drip loss	5.90	5.90	5.87	5.92	0.22	0.99	0.91
Marbling NPPC	1.52	1.46	1.50	1.48	0.06	0.65	0.86
Intramuscular fat %	2.11	2.06	2.09	2.07	0.04	0.58	0.84
Shear force (g/cm^2^)	5.30	5.65	5.42	5.30	0.12	0.58	0.77

MLOIN: minimum fat over the *Gluteus medius*. pHuLT: muscle pH at *Longissimus thoracis* 24 h post mortem (p.m.); ECuLT = electrical conductivity measured in the *Longissimus thoracis*, EJC: subjective colour using a Japanese colour scale; Marbling NPPC: subjective marbling with pattern from National Pork Producers Council (means of three assessors); L*, a*, b*: objective measure of the colour with the Minolta Chromameter. ^a^ SME: Standard error of the mean, ^b^ Housing system (HS) CE = Conventional environment/EE = enriched environment. ^c^ Herbal compound (HC) yes = supplied with herbal compound/no = not supplied. ^d^ Interaction HS*HC was not significant for any parameter.

**Table 3 animals-08-00118-t003:** Relative importance and utility values for the total of the consumers and for different clusters.

Factors and Levels	Overall	Cluster 1	Cluster 2	Cluster 3
**Feeding supplementation**				
Conventional food	−1.0	−0.7	−2.0	1.6
Supplemented food with natural herbs	1.0	0.7	2.0	−1.6
Relative importance (%)	34.4	14.3	45.9	38.4
**Production system**				
Conventional farming system	−1.3	−0.7	−1.7	−0.5
Conventional with animal welfare improvements	1.3	0.7	1.7	0.5
Relative importance (%)	45.7	13.7	40.4	13.2
**Price**				
3€	0.2	3.3	−0.5	−1.9
5€	0.5	0.6	0.7	0.0
7€	−0.6	−4.0	−0.2	2.0
Relative importance (%)	19.9	72.0	13.7	48.4
Root mean square error (RMSE)	3.0	1.3	2.2	2.6
*R*^2^	0.2	0.8	0.6	0.4

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
