# Peer review of "Effect of Environmental Enrichment and Herbal Compounds-Supplemented Diet on Pig Carcass, Meat Quality Traits, and Consumers’ Acceptability and Preference"

_animals, 2018, doi:10.3390/ani8070118_

Round 1

Reviewer 1 Report

Line 39 - remove the word also to make appropriate English

Line 43- Oxford comma needed after origin (I think Oxford commas make a lot of sense to use, if you choose to add these please do it throughout.

Line 109- remove "were used"

Line 110- Hanged needs to be hung

Line 115- describing production system as barren rather than conventional seems to be imposing your opinion on the reader. I feel our job as scientists is to present data objectively and our verbage is of the utmost importance. Please change "barren" to "conventional" throughout. I also worry that this may of influenced your survey data by indicating that the pigs in the non-enriched systems are neglected (which animal scientists know they are not)

Line 126- Typo in means listed "11.79"

Line 136- backfat is mis-spelled 
Line 155- change grinded to ground

Line 157- remove during

Line 159- units of pieces

Line 158 and Line 174- Why are the cook temperatures for panels and shear force so different. If these are reported correctly it is likely that the palatablity characteristics of the samples used for shear force are very different.

Statistics- Why use carcass weight as a covariate? I am of the opinion that carcass weight should be a response variable based on your treatments. Consider re-analysis.

Line 213- change "its" to "their"

Results- Every attribute measured must be addressed in the results section. They non-significant data can be grouped. See other published literature for examples.

Line 243- remove "different"

Please make sure Statistics accompany all reported results in the results section

Line 262- percentage does not match the table

Line 294- remove "meat". Just reference the pork as pork. Not pork meat. Please correct throughout.

Line 303- change "conflictive" to "conflicting"

Line 310- Remove "up". Also, this thought is missing your point. I think that you need to reference what your study observed for BW changes.

Lines 323 and 324- Reword, run on sentence.

Lines 325-327- please re-word to make sound more fluent. I think this may sound like a translation issue.

Line 332 and 333- End sentence after two. Delete "and found" . Capitalize Cluster

Lines 353 thru 355- Not sure this statement belongs in scientific literature. It seems to be judgemental and stereotypical. As stated before scientific literature is supposed to objectively report facts for readers to form their own educated opinion on. 

Line 357- change "ration" to "ratio"

Line 368- Change "life" to "live"

Author Response

REVISIONS ACCORDING TO REVIEWER 1

(Revisions undertaken regarding each suggestion are in bold and italics)

Line 39 - remove the word also to make appropriate English. Done

Line 43- Oxford comma needed after origin (I think Oxford commas make a lot of sense to use, if you choose to add these please do it throughout. Done, oxford commas used.

Line 109- remove "were used" Done

Line 110- Hanged needs to be hung. Done

Line 115- describing production system as barren rather than conventional seems to be imposing your opinion on the reader. I feel our job as scientists is to present data objectively and our verbage is of the utmost importance. Please change "barren" to "conventional" throughout. I also worry that this may of influenced your survey data by indicating that the pigs in the non-enriched systems are neglected (which animal scientists know they are not).

The word “barren” has been changed to conventional, except for the title of a reference in which the word “barren” was used. The authors would like to apologize because they are non-native English speakers and did not interpret that barren could have a possible misleading connotation. They actually used the term “conventional” and not barren in the cards provided to consumers in the consumer tests as seen in Table 3, since “barren” has not a straightforward word in Catalan/Spanish.

Line 126- Typo in means listed "11.79" Changed to 111.79

Line 136- backfat is mis-spelled. Corrected. 
Line 155- change grinded to ground. Changed.

Line 157- remove during. Done

Line 159- units of pieces. Added

Line 158 and Line 174- Why are the cook temperatures for panels and shear force so different. If these are reported correctly it is likely that the palatablity characteristics of the samples used for shear force are very different.

The material and methods regarding this point were maybe not sufficiently clear. The temperature for panels and shear force was actually very similar: 75 and 76ºC . However, to reach this internal temperature, the oven was pre-heated at 200ºC, and there was a mistake in the pre-heating temperature to evaluate shear force, it said 110ºC when it was 200ºC as for the shear force studies. The authors have changed the mistake accordingly.

Statistics- Why use carcass weight as a covariate? I am of the opinion that carcass weight should be a response variable based on your treatments. Consider re-analysis.

Carcass weight was actually studied as response variable (without covariate) to the treatments, as presented in Table 2, with no effect of treatments. However, the authors used carcass weight as covariate in the other variables studied to correct for differences due to different carcass weight on an individual basis

Line 213- change "its" to "their". Done

Results- Every attribute measured must be addressed in the results section. They non-significant data can be grouped. See other published literature for examples. Information added in lines 233-236)

Line 243- remove "different" Done

Please make sure Statistics accompany all reported results in the results section

Done

Line 262- percentage does not match the table. Changed

Line 294- remove "meat". Just reference the pork as pork. Not pork meat. Please correct throughout. “Meat” after pork has been removed throughout the paper.

Line 303- change "conflictive" to "conflicting”. Changed

Line 310- Remove "up". Also, this thought is missing your point. I think that you need to reference what your study observed for BW changes.

The sentence on  no previous references regarding herbal compounds provision and performance has been removed and only data published in Casal et al 2017 discussed.

Lines 323 and 324- Reword, run on sentence. “Used” has been reworded to “run” as suggested.

Lines 325-327- please re-word to make sound more fluent. I think this may sound like a translation issue. Reworded to “The overall conjoint analysis of the present study revealed that production system was the most determinant attribute, followed by feeding with herbal compound supplementation and price was the least important attribute”.

Line 332 and 333- End sentence after two. Delete "and found" . Capitalize Cluster. Done. A comma has been added before “whilst consumers from Clusters two…” to clarify the meaning of the sentence.

Lines 353 thru 355- Not sure this statement belongs in scientific literature. It seems to be judgemental and stereotypical. As stated before scientific literature is supposed to objectively report facts for readers to form their own educated opinion on. 

The statement has been removed.

Line 357- change "ration" to "ratio". Done

Line 368- Change "life" to "live". Done.

Reviewer 2 Report

The paper reports carcass, organoleptic and consumer perception results associated with a 2x2 factorial experiment to investigate stress-reduction strategies for finishing pigs. The experiment is soundly designed and analysed, and clearly reported although some improvement of English would be of benefit (some suggestions are made below). Results are of interest in building a multi-national picture of the willingness of consumers to pay for enhanced animal welfare.

L38. ‘..subject individuals..’

L39. Argued by who?

L49. ‘..with 59%..’

L51. ‘..ranged from 93% to 22%.’

L58. ‘..versus 38%..’

L71. ‘..and at reducing, ….’

L105. I think one replicate in each of 2 rooms? (2 replicates in total)

L106. ‘..adjacent pens..’

L108. ‘hoppers’

L110. ‘hung on the walls’

L113. What quantity of sawdust was provided?

L116. This is not currently legal under EU legislation.

L119. Was food available ad libitum?

L126. ‘11.79’ looks like a tying error

L145. Delete ‘equipped’

L151. Delete ‘3’

L155. ‘..ground using..’

L257. ‘..consumers, with price being the most..’

L303.’..conflicting results..’

L308. ‘Along these lines,..’

L310. ‘To the authors’ knowledge..’

L357. ‘..gender ratio..’

L358. Delete ‘with a more balanced gender ratio’ (repeated text)

L368. ‘..liveweight..’

Author Response

REVISIONS ACCORDING TO REVIEWER 2

(Revisions undertaken regarding each suggestion are in bold and italics)

The paper reports carcass, organoleptic and consumer perception results associated with a 2x2 factorial experiment to investigate stress-reduction strategies for finishing pigs. The experiment is soundly designed and analysed, and clearly reported although some improvement of English would be of benefit (some suggestions are made below). Results are of interest in building a multi-national picture of the willingness of consumers to pay for enhanced animal welfare.

L38. ‘..subject individuals..’ Changed

L39. Argued by who?

A reference has been added Harper, G.; Henson, S. Consumer concerns about animal welfare and the impact on food choice. EU FAIR CT98-3678. Final Report. 2001, Centre for Food Economics Research, University of Reading, UK.

L49. ‘..with 59%..’ changed

L51. ‘..ranged from 93% to 22%.’ changed

L58. ‘..versus 38%..’changed

L71. ‘..and at reducing, ….’ changed

L105. I think one replicate in each of 2 rooms? (2 replicates in total). It has been reworded.

L106. ‘..adjacent pens..’ changed

L108. ‘hoppers’ changed

L110. ‘hung on the walls’ changed

L113. What quantity of sawdust was provided?

Two full buckets of 50 x 60 cm every two days. This information has been added

L116. This is not currently legal under EU legislation.

This issue has been addressed with the editor (two revision notes sent) and the following paragraph added at the beginning of the material and methods section:

The housing, husbandry and use of the animals for the procedures described in this manuscript were carried out according to the European legislation for animals used for scientific purposes (Directive 2010/63/EU of the EU Parliament and of the Council of 22 September 2010 on the protection of animals used for scientific purposes). The project, including this experimental procedure, was approved by IRTA’s (Institute of Agrifood Research and Technology) Ethical Committee. According to Directive 2010/63/EU, “any restrictions on the extent to which an animal can satisfy its physiological and ethological needs have to be kept to a minimum”. In the present experiment, no environmental enrichment was provided to the control group, because the authors aimed at investigating the effects of the provision of enrichment on behavioural, physiological, performance and meat and carcass quality traits. For that reason, the Ethical Committee had to decide according to Article 38 of EU legislation on whether “the harm to the animals in terms of suffering, pain and distress is justified by the expected outcome taking into account ethical considerations, and may ultimately benefit human beings, animals or the environment”. The Ethical Committee provided its approval to the procedure considering that the data obtained regarding provision of enrichment would yield important outcomes to be implemented on commercial farms, for which, according to EU legislation (EU Council directive on pig welfare 2008/120/EC), enrichment material is mandatory.  In order to minimize the possible effects of non provision of enrichment to the control group, other risk factors which can influence the fulfilment of pig behavioral needs and, as a consequence, on tail biting, were cared for: space allowance per pig was 1.95 m2/animal; climatic conditions were automatically controlled and daily supervision was carried out according to the Ethical Committee recommendations to apply correction measures when appropriate.

L119. Was food available ad libitum?

Yes, this information has been added

L126. ‘11.79’ looks like a tying error

It has been changed to 111.79

L145. Delete ‘equipped’

Done

L151. Delete ‘3’

Done

L155. ‘..ground using..’

Changed

L257. ‘..consumers, with price being the most..’

Changed

L303.’..conflicting results..’

Changed

L308. ‘Along these lines,..’

Changed

L310. ‘To the authors’ knowledge..’

Sentence removed to answer reviewer 1 comments

L357. ‘..gender ratio..’

Changed

L358. Delete ‘with a more balanced gender ratio’ (repeated text)

Deleted

L368. ‘..liveweight..’

Changed

Round 2

Reviewer 1 Report

Thank you for taking into account my edits. Good luck in the remainder of the publication process.

Reviewer 2 Report

The authors have dealt satisfactorily with all points raised in my earlier review.